# Estimating relative CWD susceptibility and disease progression in farmed white-tailed deer with rare PRNP alleles

**Nicholas J. Haley** [1]*, **Kahla Merrett**[1], **Amy Buros Stein**[2], **Dennis Simpson**[3], **Andrew Carlson**[3], **Gordon Mitchell**[4], **Antanas Staskevicius**[4], **Tracy Nichols**[5], **Aaron D. Lehmkuhl**[6], **Bruce V. Thomsen**[6,7]

**1** Department of Microbiology and Immunology, College of Graduate Studies, Midwestern University, Glendale, Arizona, **2** Office of Research and Sponsored Programs, Midwestern University, Glendale, Arizona, **3** Simpson Whitetails Genetic Testing, Belleville, Michigan, **4** National and OIE Reference Laboratory for Scrapie and CWD, Canadian Food Inspection Agency, Ottawa Laboratory-Fallowfield, Ottawa, Ontario, Canada, **5** United States Department of Agriculture, APHIS, Veterinary Services, Cervid Health Program, Fort Collins, Colorado, United States of America, **6** United States Department of Agriculture, APHIS, Veterinary Services, National Veterinary Services Laboratories, Ames, Iowa, United States of America, **7** United States Department of Agriculture, APHIS, Veterinary Services, Center for Veterinary Biologics, Ames, Iowa, United States of America

* NHaley@midwestern.edu

**Data Availability Statement:** All relevant data are within the paper and its Supporting Information files.

## Abstract

Chronic wasting disease is a prion disease affecting both free-ranging and farmed cervids in North America and Scandinavia. A range of cervid species have been found to be susceptible, each with variations in the gene for the normal prion protein, PRNP, reportedly influencing both disease susceptibility and progression in the respective hosts. Despite the finding of several different PRNP alleles in white-tailed deer, the majority of past research has focused on two of the more common alleles identified—the 96G and 96S alleles. In the present study, we evaluate both infection status and disease stage in nearly 2100 farmed deer depopulated in the United States and Canada, including 714 CWD-positive deer and correlate our findings with PRNP genotype, including the more rare 95H, 116G, and 226K alleles. We found significant differences in either likelihood of being found infected or disease stage (and in many cases both) at the time of depopulation in all genotypes present, relative to the most common 96GG genotype. Despite high prevalence in many of the herds examined, infection was not found in several of the reported genotypes. These findings suggest that additional research is necessary to more properly define the role that these genotypes may play in managing CWD in both farmed and free-ranging white-tailed deer, with consideration for factors including relative fitness levels, incubation periods, and the kinetics of shedding in animals with these rare genotypes.

## Introduction

Chronic wasting disease (CWD) is a progressive neurologic disease of cervids caused by a transmissible, misfolded protein—the prion protein. [1,2] The disease is naturally occurring in

**Funding:** This work was supported by the North American Deer Farmers Association (https://nadefa.org/) to NH and Whitetails of Wisconsin (https://www.whitetailsofwisconsin.com/) to NH. Simpson Whitetails Genetic Testing provided support in the form of salaries for authors (DS and AC), as well as genotypic data from a subset of CWD negative herds, representing approximately 50% of the healthy herds reported. These authors had no role in study design or data analysis. The specific roles of these authors are further articulated in the "author contributions" section.

**Competing interests:** Two authors (DS and AC) were employed by Simpsons Whitetails Genetic Testing at the time this study was undertaken. These two authors provide a commercial test for PRNP genotyping to the deer and elk farming industry. There are no patents or products in development associated with this commercial venture. This association does not alter our adherence to PLOS ONE policies on sharing data and materials.

white-tailed deer (*Odocoileus virginianus*), mule deer (*Odocoileus hemionus*), Rocky Mountain elk and red deer (*Cervus elaphus* sspp.), moose (*Alces alces*), and reindeer (*Rangifer tarandus*). [3] It has been reported in farmed and free-ranging cervids in 26 US states, 3 Canadian provinces, the Republic of Korea, Norway, Sweden, and Finland. [3–8] Chronic wasting disease is highly transmissible through direct contact or environmental contamination, and has been detected at various levels in all tissues and bodily fluids of cervids examined to date. [9–18]

The misfolded protein—commonly designated "PrP$^{res}$" due to its resistance to harsh physical treatments, is derived from the cellular prion protein—"PrP$^C$"—a normal protein encoded by the *PRNP* gene, which is present in a range of animals in the phylum Chordata. [19–21] Prion disease transmission and pathogenesis relies on the coerced conversion of normal PrP$^C$ by PrP$^{res}$ into the abnormally folded isoform, which collects in the form of amyloid in a variety of tissues, most notably the central nervous system, resulting in the eventual demise of the host. The tertiary structure of this misfolded protein, and its properly folded counterpart, is inherently dependent on its primary amino acid sequence. [22–30] As such, the ability of the misfolded protein to coerce normally folded prion proteins into an abnormal, amyloid-forming structure is highly dependent on the primary amino acid sequence of both the infectious and host prion proteins. Significant variation between host and infectious prion proteins results in reduced host susceptibility, and in some cases complete resistance to disease—a phenomenon known as the "species barrier" when considering natural or experimental inter-species transmission of the infectious prion agent. [31–35]

Variations in prion disease susceptibility have been reported across most species naturally affected by these agents. Humans with variation in amino acids at either position 127 or 129 are resistant to various transmissible forms of Creutzfeldt-Jakob disease and Kuru. [36] Goats with amino acid variations present at positions 146, 211, and 222, as well as several other sites, show reduced susceptibility to either BSE or sheep scrapie. [24,37,38] Sheep with variations at position 136, 154, and 171, among others, present with a range of susceptibilities to classical scrapie—including, in the case of A$_{136}$R$_{154}$R$_{171}$ homozygous sheep, near-complete resistance to infection. [39–41] The latter finding has led to a multinational effort to breed sheep towards resistance to classical scrapie infection in areas where the disease is endemic, resulting in a significant decline and near-eradication of the disease in countries employing targeted breeding programs. [42–44]

Polymorphisms in the *PRNP* gene of white-tailed deer, mule deer, elk, fallow deer and reindeer have all been found to influence susceptibility to CWD in wild, farmed, and experimental populations. [26,45–49] The low prevalence of CWD in these populations has often made it difficult to adequately understand the role these polymorphisms may play in the disease process. Additionally, many of these studies only incorporate a binary (positive or not detected) approach to disease diagnosis, and fail to include disease staging as a factor in susceptibility. [45,50–53] Lastly, and perhaps most importantly, most of these polymorphisms are quite rare, and animals homozygous for these alleles, or in rare heterozygous combinations, have neither been observed in CWD endemic populations nor tested for their susceptibility following natural exposure. [26] Exceptions include the 225F polymorphism in mule deer and the 132L polymorphism in Rocky Mountain elk. In the case of 225FF homozygous mule deer, a small group of animals placed on a heavily contaminated pasture eventually developed progressive neurologic disease and neuropathology characteristic of CWD, although the 225F allele seems to be a significant barrier to infection in wild populations under more typical exposure conditions. [45,54] Elk heterozygous or homozygous for the 132L polymorphism likewise show reduced susceptibility in both wild and captive populations, however 132LL homozygous elk have only rarely been found to be infected. [48,55]

In the present study, we sought to better define the relative susceptibilities of white-tailed deer heterozygous and homozygous for several different *PRNP* alleles, including 95H, 96G and 96S, 116G, and 226K. Samples were analyzed from nearly 2100 farmed deer depopulated following exposure to CWD, including 714 deer infected with CWD, with postmortem prevalence ranging from 6–83% across 20 separate herds in the United States and Canada. In addition to CWD status, we also examined the correlation of *PRNP* genotype with the stage of disease, ranging from one (detection in retropharyngeal lymph nodes, RLN, only) to five (detection in RLN in addition to significant immunostaining in the obex region of the brainstem). Finally, we surveyed 117 healthy white-tailed deer herds in the United States and 7 white-tailed deer herds in Canada to assess the distribution of these five different alleles across North American farmed deer populations. We hypothesized that CWD status and disease stage would be most significant and severe in animals homozygous for the 96G allele, with other pairings less significantly and severely affected. We also hypothesized that allele frequencies would vary between the Canada and the United States, and within geographic regions of the United States. We found several combinations of alleles that were associated with significantly reduced CWD prevalence and/or disease severity, and that specific alleles may be more common in different regions of North America. These findings suggest that variations in susceptibility to CWD may play a role in managing the disease in this species, and those variations may be more common in farmed deer in certain areas, warranting further exploration of *PRNP* markers in the natural white-tailed deer host.

## Methods

### Ethics statement

This study was conducted retrospectively, using tissues collected from animals depopulated during the normal course of disease control efforts by the United States Department of Agriculture and the Canadian Food Inspection Agency. Sequencing of *PRNP* genotypes in healthy animals was conducted on a contract testing basis, with results anonymized and with the consent of the owners submitting the samples.

### Study population

Twenty white-tailed deer herds depopulated in the United States (11 herds with 1185 adult animals) and Canada (9 herds with 906 adult animals) were included in the analysis. Each herd had initially reported one or more deer with a positive diagnosis of CWD, and was subsequently placed under quarantine. When the animals were later depopulated, a variety of samples were collected, including RLN and the obex region of the brainstem for conventional CWD testing, and either blood or ear punch for PCR amplification and sequencing of the *PRNP* gene. All herds were depopulated in roughly the past 5–10 years, though not all herds depopulated in those years had samples available, and in some cases samples were not available from all animals in their respective herds. Missing samples included animals too young to test, animals from which a sample was otherwise unavailable, and animals with poor quality DNA samples. Animal age was generally unknown, though only adult animals over 1 year of age were considered for the study. Of the 2091 animals evaluated, 714 were ultimately found to be CWD positive (34.1%). Further details on the herd sizes, their country of origin, gene frequencies, and CWD prevalence (based on results from cases with available DNA) can be found in Tables 1 and 2.

**Table 1. Summary of herds in the United States providing samples for the present study.** Eleven herds in the United States, comprised of 1185 samples from individual deer were included in the analysis. Prevalence and genotype data from each herd, based on animals for which both genetic data and CWD status are available, are shown.

| Herd ID | Number present | Number available for testing | CWD Prevalence (%) | Genotype | | | | | | | | | | | | | | | |
| | | | | 96GG | | 96GS | | 96SS | | 95H/96G | | 95H/96S | | 95HH | | 96G/226K | | 96S/226K | |
| | | | | - | + | - | + | - | + | - | + | - | + | - | + | - | + | - | + |
| A | 81 | 80 | 12.5 | 36 | 7 | 22 | 3 | 5 | 0 | 0 | 0 | 0 | 0 | 0 | 0 | 6 | 0 | 1 | 0 |
| B | 99 | 96 | 9.4 | 30 | 4 | 38 | 3 | 12 | 1 | 1 | 0 | 0 | 0 | 0 | 0 | 5 | 1 | 1 | 0 |
| C | 47 | 47 | 12.8 | 19 | 4 | 15 | 2 | 7 | 0 | 0 | 0 | 0 | 0 | 0 | 0 | 0 | 0 | 0 | 0 |
| D | 140 | 140 | 5.7 | 68 | 8 | 41 | 0 | 12 | 0 | 5 | 0 | 3 | 0 | 0 | 0 | 2 | 0 | 1 | 0 |
| E | 129 | 128 | 19.5 | 44 | 23 | 33 | 1 | 6 | 0 | 11 | 0 | 0 | 0 | 2 | 0 | 5 | 1 | 2 | 0 |
| F | 99 | 99 | 9.1 | 60 | 3 | 25 | 6 | 3 | 0 | 0 | 0 | 0 | 0 | 0 | 0 | 2 | 0 | 0 | 0 |
| G | 85 | 79 | 26.6 | 51 | 20 | 7 | 1 | 0 | 0 | 0 | 0 | 0 | 0 | 0 | 0 | 0 | 0 | 0 | 0 |
| H | 116 | 116 | 22.4 | 25 | 20 | 41 | 5 | 21 | 0 | 0 | 0 | 0 | 0 | 0 | 0 | 2 | 1 | 1 | 0 |
| I | 18 | 14 | 35.7 | 2 | 5 | 5 | 0 | 0 | 0 | 0 | 0 | 1 | 0 | 0 | 0 | 1 | 0 | 0 | 0 |
| J | 356 | 356 | 79.8 | 7 | 120 | 37 | 138 | 24 | 19 | 3 | 5 | 0 | 0 | 0 | 0 | 0 | 2 | 1 | 0 |
| K | 36 | 30 | 20.0 | 16 | 4 | 5 | 2 | 1 | 0 | 1 | 0 | 0 | 0 | 0 | 0 | 0 | 0 | 1 | 0 |
| **Total** | **1206** | **1185** | **34.5** | **358** | **218** | **269** | **161** | **91** | **20** | **21** | **5** | **4** | **0** | **2** | **0** | **23** | **5** | **8** | **0** |

## Unaffected populations

Samples from healthy animals in 117 herds in the United States (n = 6030 animals) and 7 herds in Canada (n = 1313 animals) were also evaluated for *PRNP* genotype frequencies. Herds in the United States were further subcategorized by region, and included 75 herds from the Midwest (n = 3865 animals tested from Iowa, Indiana, Michigan, Minnesota, Missouri, North Dakota, Ohio and Wisconsin), 29 from the Northeast (n = 1651 from Pennsylvania), and 13 from the South (n = 514 from Texas and Alabama). Although it was common for entire herds to be included in the analysis, it is important to note that herds submitting samples for testing were not likely to be random and the number of states, and herds included, in each region varied. A summary of allele frequencies in white-tailed deer herds in the United States and Canada may be found in Table 3. Genotypic data are also provided in S1 Table.

**Table 2. Summary of herds in Canada providing samples for the present study.** Nine herds from Canada, comprised of 906 samples from individual deer, were included in the analysis. Prevalence and genotype data from each herd, based on animals for which both genetic data and CWD status are available, are shown.

| Herd ID | Number present | Number available for testing | CWD Prevalence (%) | Genotype | | | | | | | | | | | | | | |
| | | | | 96GG | | 96GS | | 96SS | | 95H/96G | | 96G/116G | | 96S/116G | | 116GG | |
| | | | | - | + | - | + | - | + | - | + | - | + | - | + | - | + |
| L | 72 | 43 | 30.2 | 15 | 11 | 7 | 2 | 0 | 0 | 0 | 0 | 6 | 0 | 0 | 0 | 2 | 0 |
| M | 29 | 29 | 82.8 | 0 | 11 | 4 | 12 | 1 | 1 | 0 | 0 | 0 | 0 | 0 | 0 | 0 | 0 |
| N | 56 | 55 | 23.6 | 16 | 8 | 18 | 1 | 3 | 0 | 0 | 0 | 4 | 4 | 1 | 0 | 0 | 0 |
| O | 179 | 133 | 8.3 | 67 | 8 | 32 | 3 | 9 | 0 | 0 | 0 | 11 | 0 | 3 | 0 | 0 | 0 |
| P | 325 | 241 | 58.1 | 28 | 82 | 45 | 36 | 6 | 2 | 0 | 1 | 13 | 17 | 8 | 2 | 1 | 0 |
| Q | 23 | 12 | 41.7 | 3 | 3 | 2 | 1 | 1 | 0 | 0 | 0 | 1 | 1 | 0 | 0 | 0 | 0 |
| R | 70 | 47 | 63.9 | 0 | 9 | 10 | 19 | 6 | 2 | 0 | 0 | 1 | 0 | 0 | 0 | 0 | 0 |
| S | 66 | 35 | 11.4 | 10 | 1 | 12 | 1 | 1 | 0 | 0 | 1 | 7 | 1 | 0 | 0 | 1 | 0 |
| T | 414 | 311 | 20.9 | 110 | 47 | 6 | 1 | 0 | 0 | 0 | 0 | 106 | 17 | 8 | 0 | 16 | 0 |
| **Total** | **1264** | **906** | **33.8** | **249** | **180** | **136** | **76** | **27** | **5** | **0** | **2** | **149** | **40** | **20** | **2** | **20** | **0** |

**Table 3. Summary of genotype frequencies in healthy North American white-tailed deer herds.** Data from whole herds opting to perform PRNP genotyping were included in the analysis, which found significant differences in distribution between Canada and the United States, as well as between specific regions of the United States.

| Location | Number of Herds | Number of Animals | Allele Frequency % | | | | |
|---|---|---|---|---|---|---|---|
| | | | 95H | 96G | 96S | 116G | 226K |
| United States | | | | | | | |
| Midwest | 75 | 3865 | 1.5 | 72.6 | 22.1 | 0 | 3.6 |
| Northeast | 29 | 1651 | 3.1 | 71.5 | 21.1 | 0 | 4.1 |
| South | 13 | 514 | 0 | 58.1 | 39.2 | 0 | 2.7 |
| **United States Total** | **117** | **6030** | **1.8** | **71.0** | **23.3** | **0** | **3.7** |
| Canada | | | | | | | |
| Alberta | 4 | 629 | 0.56 | 67.1 | 29.8 | 2.5 | 0 |
| Saskatchewan | 2 | 684 | 2.2 | 62.9 | 31.4 | 3.3 | 0 |
| **Canada Total** | **6** | **1313** | **1.4** | **65.0** | **30.7** | **2.9** | **0** |

### *PRNP* analysis

For CWD correlation, nucleic acids were extracted in most cases from whole blood samples preserved in EDTA, or in some cases ear punch biopsies, using a conventional DNA extraction kit. (ThermoFisher, USA) For healthy herd gene frequencies, DNA was most commonly extracted from hair samples provided by healthy herds across North America, though semen, antler core, ear notches and other biopsy samples were also included. Data from these healthy herds solely included locations where the entire herd was sampled. An approximately 750bp *PRNP* gene sequence was amplified by conventional PCR and sequenced as previously described. [46,56] PCR sequences were aligned and evaluated using Geneious software version 10.2 (www.Geneious.com). Specific single nucleotide polymorphisms at position 95 (glutamine [Q] or histidine [H]), 96 (glycine [G] or serine [S]), 116 (alanine [A] or glycine), and 226 (glutamine or lysine [K]) were identified and recorded.

### Immunohistochemistry of retropharyngeal lymph node and brainstem

Retropharyngeal lymph node and brainstem tissues were examined microscopically for PrP$^{CWD}$ immunostaining as previously described. [56,57] Briefly, tissue was preserved in 10% neutral buffered formalin and subsequently embedded in paraffin blocks. Tissue sections 5 μm thick were mounted on glass slides and deparaffinized before treatment with 95% formic acid. Immunohistochemical staining for PrP$^{CWD}$ was performed with the primary antibody anti-prion 99 (Ventana Medical Systems, Tucson, AZ) and then counterstained with hematoxylin. The obex sections were scored from 0 to 4 on the basis of the following criteria: grade 0, no IHC staining observed within the obex; grade 1, IHC staining only within the dorsal motor nucleus of the vagus (DMNV); grade 2, IHC staining within the DMNV and area postrema with or without focal staining in the nucleus of the solitary tract (NST) and adjacent white matter; grade 3, IHC staining in the DMNV and NST with light to moderate staining extending into other nuclei and white matter; grade 4, heavy IHC staining of the DMNV, multiple other nuclei, and white matter throughout the obex. Results were tabulated according to RLN and obex immunostaining, with individuals exhibiting immunostaining in the RLN alone scored as a "1," while those with additional immunostaining in the obex scored as 2–5 depending on obex staining intensity. As with previous studies, all deer that had obex staining always concurrently had staining in the RLN, a finding characteristic of CWD in white-tailed deer.

## Statistical analyses

Statistical analysis was done using R version 3.5.1 with the *brms* [58] and *nlme* [59] packages. A linear mixed model, with herd included as a random effect, was used to calculate coefficients for disease stages relative to the 96GG genotype with associated 95% confidence intervals. A Bayesian mixed effects logistic regression model with herd again included as a random effect was used to determine odds ratios of infection in various genotypes relative to the 96GG genotype. A weakly informative prior for genotypes was defined as the Cauchy distribution with location and scale parameters of 0 and 2.5, respectively. This approach accounted for differences in disease prevalence and genotypic distribution between and across farms in an effort to better estimate relative susceptibility and disease progression. The Markov-chain Monte-Carlo (MCMC) sampling was used with 500000 iterations, following an initial burn-in period of 5000 iterations. The scale reduction factor was calculated to assess convergence and adequate mixing of the chains. The posterior medians and 95% credible intervals were used for inference.

In order to predict outcomes for genotypes that were not observed, an additive mixed effects model, both linear and logistic, were built using data from measured allele pairs to estimate the contribution of each single allele. The prediction interval for the log odds estimate was calculated using the *merTools* package [60] and is done by drawing a sampling distribution for the random and fixed effects and then estimating the fitted value across that distribution. The calculated interval includes all variation in the model except for variation in the covariance parameters.

A chi-squared test was used to compare *PRNP* frequencies between Canada and the United States, as well as between different regions of the United States.

## Results

### Correlation of *PRNP* genotype with CWD infection status

Positive and negative CWD infection status were correlated to *PRNP* genotypes using the 96GG genotype as a reference point to assess odds ratios of infection. A significant reduction in odds ratio of infection was seen with all genotypes examined, except for the 96G/226K genotype. While there was a trend towards reduced odds ratios in this genotype, the findings were not statistically significant. Among animals heterozygous for the 96G allele, odds ratios were lowest in animals carrying the 95H allele (0.257, 95% CI: 0.08–0.80), though the results were not significantly different than those found in animals with the 96GS genotype (0.319, 95% CI: 0.23–0.43). Among alleles for which sufficient data were available for modeling, animals homozygous for the 116G allele had the lowest odds ratio of being found infected ($3 \times 10^{-6}$), though confidence intervals ranged widely. Results are summarized in Table 4 and Fig 1. Modeling odds ratios of infection in non-96G homozygous genotypes continued to exhibit wide-ranging confidence intervals, though suggested that 95HH homozygous genotypes in particular may have the lowest odds ratios for being found CWD positive (Fig 2).

### Correlation of *PRNP* genotype with CWD infection stage

Disease stages were correlated to rare *PRNP* genotypes, again using the 96GG genotype as a reference point to evaluate differences in disease severity. In all genotypes examined, a significant reduction in disease staging was observed compared to the 96GG reference genotype. As noted with odds ratios above, the most significant reduction in disease staging was observed in animals with the 95H/96G genotype (-1.205, 95% CI: -1.66 to -0.75), though again this finding

**Table 4. Relative CWD susceptibility and disease staging in white-tailed deer with rare alleles, in reference to the 96GG genotype.** Odds ratio of identifying infection in rare alleles was determined using Bayesian mixed effects logistic regression, while relative disease stages were calculated using linear coefficient modeling. Significantly lower odds of being found infected, relative to the 96GG genotype, were observed in all rare genotypes except for the 96G/226K genotype, where findings were suggestive of lower odds ratios, though statistically inconclusive. Negative values for disease staging indicate a trend towards earlier stages of disease, and a significantly lower disease stage was found in all rare genotypes evaluated relative to animals with the 96GG genotype.

| Genotype | Bayes Logistic OR | Logistic 95% CI | Linear Coefficient | Linear 95% CI |
|---|---|---|---|---|
| 96GS | 0.319 | (0.23, 0.43) | -0.839 | (-0.96, -0.72) |
| 96SS | 0.069 | (0.04, 0.12) | -1.502 | (-1.72, -1.29) |
| 95H/96G | 0.257 | (0.08, 0.80) | -1.205 | (-1.66, -0.75) |
| 96G/116G | 0.440 | (0.28, 0.68) | -0.463 | (-0.67, -0.26) |
| 96G/226K | 0.551 | (0.18, 1.39) | -0.828 | (-1.28, -0.38) |
| 96S/116G | 0.090 | (0.02, 0.36) | -1.130 | (-1.63, -0.63) |
| 116GG | 0.000003 | (0.00, 0.30) | -0.853 | (-1.39, -0.32) |
| 96S/226K | 0.00005 | (0.00, 0.68) | -1.137 | (-1.96, -0.31) |
| 95H/96S | 0.018 | (0.00, 2.56) | -0.744 | (-1.92, 0.43) |

was not significantly different than what was observed for 96GS heterozygous animals (-0.839, 95% CI: -0.96 to -0.72). Among homozygous animals with sufficient data available for modeling, disease staging was lowest in animals with the 96SS genotype, though it should be noted that low or absent numbers of rarer genotypes made their analysis challenging. Results again are summarized in Table 4 and Fig 1, with models addressing disease progression in other homozygous genotypes again presented in Fig 2.

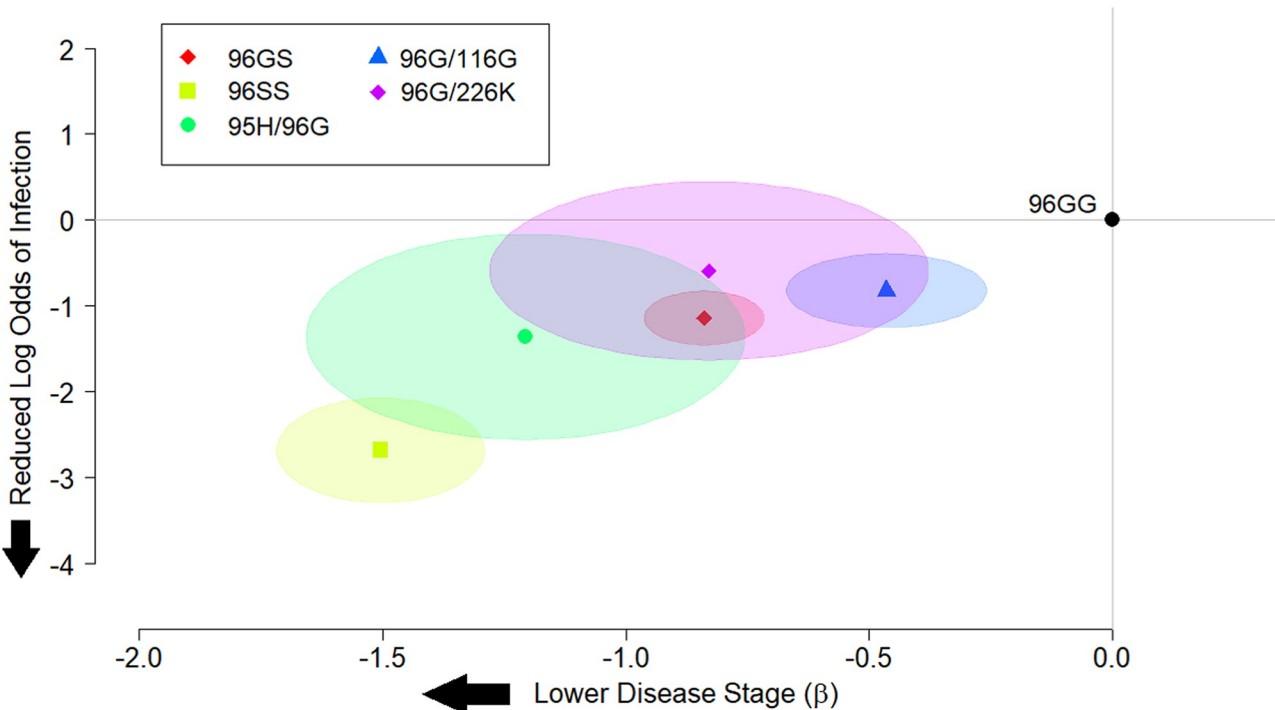

**Fig 1. Summary of log odds ratios of white-tailed deer with 96G heterozygous and 96SS homozygous genotypes being found CWD positive, and the stage of disease recorded among those infected relative to the 96GG genotype.** The most common genotypes found in the study are presented, showing that all heterozygous 96G crosses exhibit some level of slowed disease progression and/or reduced susceptibility.

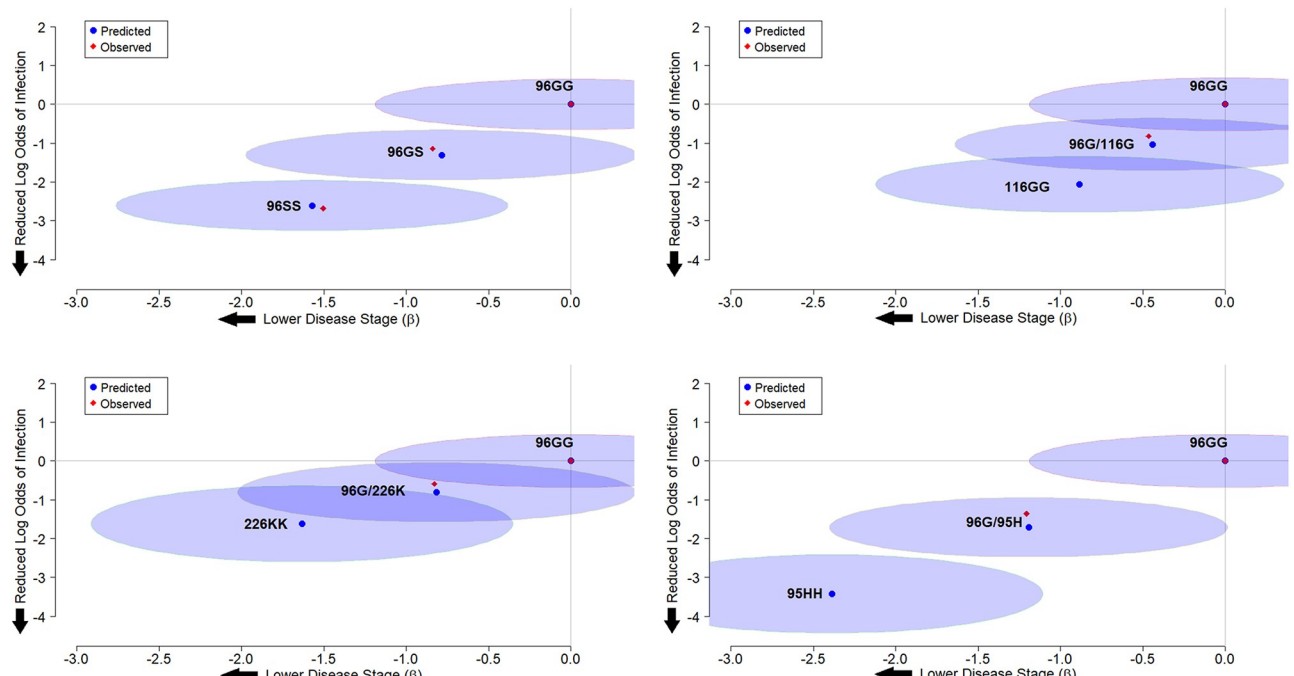

**Fig 2. Estimates of log odds ratios and disease staging for the 96S, 116G, 226K, and 95H alleles in the homozygous state.** Using data from measured allele pairs, an additive mixed effects model was developed to predict outcomes in genotypes with insufficient data. Predicted estimates for disease susceptibility and progression are show for both heterozygous 96G genotypes and homozygous pairings.

### Frequency of *PRNP* alleles in healthy farmed white-tailed deer herds

Significant differences were observed in the frequency of various alleles in Canadian and US herds—particularly with regard to the exclusive presence of the 116G allele in Canadian herds and the 226K allele in US herds. The 96G allele was found to be at a significantly higher frequency in US herds, while the 96S allele was found to be at significantly higher frequencies in Canadian herds. Within the United States, significant differences in *PRNP* frequencies were also observed between different regions of the country, regions that are admittedly arbitrary with samples available only from some states within those regions. Most notably, the 95H allele was significantly more common in herds in the Northeast compared to both Midwestern and Southern herds, while the 96S allele was found at a higher frequency in Southern states compared to herds in the Midwest and Northeast. No differences in allele frequencies were observed between herds in the Canadian provinces of Alberta and Saskatchewan. (Table 3 and S1 Table)

Because the samples from healthy deer herds in both the United States and Canada are presumed to have been submitted non-randomly—e.g. those herds financially capable of testing, and those having a particular interest in *PRNP* genotyping, it is important to note that these findings should be interpreted with caution.

## Discussion

A significant amount of research over the past two decades has been conducted on *PRNP* gene frequencies in both wild and farmed white-tailed deer populations affected by CWD, which cumulatively has led to the understanding that animals with different *PRNP* alleles are differentially susceptible to CWD infection. [8,26,50,51,57] Recent research has pointed to slower

disease progression in animals with several of the more common genotypes, notably those carrying the 96S allele, in addition to their reduced susceptibility. [56,57] Each of these previous studies, however, have suffered from limitations which may hinder broader interpretation, including low disease prevalence and/or negligible or absent populations of animals representing rarer genotypes. [50–52,61] The present study represents one of the largest in-depth evaluations of the relationship between *PRNP* genotype and both CWD status and disease stage in white-tailed deer, and the relatively high disease prevalence in many of these populations provided us with important insight into susceptibility and disease progression in some of the more rare genotypes.

Previous studies have typically focused on two of the most common alleles—commonly referred to as the 96G and 96S alleles, and the corresponding 96GG, 96GS, and 96SS genotypes. Occasionally these studies make use of genotyping strategies that might ignore the contribution of other, rarer alleles. [57,62] In the present study, as in past studies, the 96G and 96S alleles made up a substantial percentage of total alleles in a population, making statistical comparisons easier even with small population sizes. [26,56] We found that, in line with previous studies, animals with the 96GS and 96SS genotypes were at a significantly reduced risk of being found CWD positive at the time of depopulation, and were generally in a significantly earlier stage of disease when infected compared to animals with the 96GG genotype.

We extended our analyses to rarer alleles, including the 95H, 116G, and 226K alleles, which to date have only garnered passing interest in susceptibility studies. [46,50,61,63] We report that the animals evaluated in this study with the 95H/96G and 96G/116G genotypes not only appear to face significantly lower risk of being found CWD positive, they, like their 96GS and 96SS counterparts, were also found to be in significantly earlier stages of disease at the time of depopulation. While there was a trend towards reduced susceptibility in animals with the 96G/226K genotype, their differences compared to animals with the 96GG genotype were not statistically significant. The 96G/226K genotype was, however, found to correlate with significantly lower disease scores than 96GG homozygous animals in the study. Models extending available data to 95HH, 116GG, and 226KK homozygous genotypes suggest the potential for an even further reduction in both susceptibility and disease progression.

To a limited extent, both the 95H and 116G alleles have been evaluated in prior studies for CWD susceptibility in either free-ranging or farmed white-tailed deer herds. A study of a wild deer population in Illinois found that animals with the 95H allele faced a risk of being found CWD-positive 1/5th that of the herd at large, similar to data reported in the present study (OR = 0.257, Table 4). [51] A limited bioassay study including two animals with the 95H allele found that CWD incubation periods were nearly double that of their 96GG and 96GS counterparts. [61] Subsequent examinations of animals in that report suggested differences in CWD prion protease sensitivity which might affect diagnostic test results—an important factor to consider when evaluating the results from the present study. [64] An evaluation of a farmed herd in Nebraska, meanwhile, found that white-tailed deer with the 116G allele were roughly half as likely to be found CWD positive compared to the herd at large, again very similar to the results reported here (OR = 0.440, Table 4). [46] Little information is available regarding the 226K allele in the natural host; however, *in vitro* misfolding studies have shown that, like several other rare cervid *PRNP* alleles, recombinant 226K prion protein is significantly limited in its ability to misfold in the presence of CWD prions. [65] Additional work is needed to more adequately define relative infection odds ratio and disease staging in not only the 96G/226K genotype, but other rare alleles as well—especially in animals homozygous for 95H, 116G, or 226K alleles.

While our findings, and those of past research efforts, suggest that deer with specific alleles face a significantly lower risk of being found CWD positive at depopulation—as well

as a significant deceleration in disease progression when infected—it seems likely that deer carrying these alleles are not completely resistant to the disease. It is therefore uncertain what role, if any, *PRNP* genetics may play in the management of CWD in both farmed and free-ranging deer. From a diagnostic perspective, animals with more susceptible alleles exhibit a more rapid progression of the disease, and are thus more readily identified on ante-mortem testing. This particular factor may prove helpful in more quickly identifying infected herds and placing them under quarantine. [56,57] The increased diagnostic sensitivity offered by animals with susceptible genotypes, however, should be carefully weighed against the drawbacks of raising highly susceptible animals, especially in areas where CWD is highly endemic.

Apart from the diagnostic challenges noted above, additional factors that should be considered include the role that less susceptible alleles may have on general animal health, any delays in disease progression, and the resultant kinetics of prion shedding in infected animals carrying them. At present, there is almost no objective information available on the fitness of various *PRNP* genotypes in cervids [54], and while there are several limited reports of CWD prion shedding in more common white-tailed deer genotypes [66–68], the biological relevance of prions likely shed in biological fluids has proven more difficult to assess. [10,15,69] The lifespan of the host is also relevant when considering incubation periods of the disease—particularly in farmed deer, where age may be useful as a selective management factor, similar to strategies used to address concerns for zoonotic transmission of BSE from cattle. [70] Lastly, it is critical to understand the mutable nature of the CWD prion agent itself in the face of shifting host genetic background, and whether any novel strains that may arise have any notable differences in disease manifestation and zoonotic potential. [71–74]

In free-ranging herds, it is even less clear if there is a role for human intervention, and more importantly whether CWD may be actively shaping *PRNP* allele frequencies in wild populations. [26] At least one study has found that the less susceptible 96S allele may provide a significant fitness advantage in a CWD endemic area, making it especially valuable to reevaluate the current frequencies of *PRNP* alleles in areas hard hit by the CWD epidemic. [62] As with farmed deer, understanding the relationship between *PRNP* genotype, fitness, prion shedding, and incubation periods would prove useful to those seeking to manage the disease in wild herds as allele frequencies shift over time. Our surveillance efforts in farmed populations shows that rare alleles are fairly well distributed across North America, with potential regional variation in frequencies, and similar efforts in wild cervids in both North America and Scandinavia may prove both useful and informative.

In summary, we provide further evidence that specific and often rare *PRNP* alleles of white-tailed deer appear to correlate strongly to both CWD susceptibility and progression. Though rare, these alleles may be found in farmed deer herds across the United States and Canada, with potential, as yet unexplained, regional variations observed. Ongoing studies in farmed deer should provide some insight into both the relative fitness of animals carrying these alleles and their utility in managing CWD in endemic areas. The role these genotypes may have in managing the disease in free-ranging white-tailed deer should likewise continue to be explored, within the context of those considerations noted above.

## Supporting information

**S1 Table. Alternative presentation of Table 2, to present genotype data from healthy US and Canadian white-tailed deer herds.**
(DOCX)

## Acknowledgments

This study was funded in large part by the North American Deer Farmers Association and Whitetails of Wisconsin. The authors would like to thank all of the deer farmers who were involved in the depopulation events, which formed the basis for the present study. We are also thankful for the support personnel working with the United States Department of Agriculture, the National Veterinary Services Laboratory, and the Canadian Food Inspection Agency who helped facilitate post-mortem sample collection and processing for this study.

## Author Contributions

**Conceptualization:** Nicholas J. Haley.

**Data curation:** Nicholas J. Haley, Kahla Merrett, Dennis Simpson, Andrew Carlson.

**Formal analysis:** Nicholas J. Haley, Kahla Merrett, Amy Buros Stein.

**Funding acquisition:** Nicholas J. Haley, Dennis Simpson.

**Investigation:** Nicholas J. Haley, Kahla Merrett, Gordon Mitchell, Antanas Staskevicius, Aaron D. Lehmkuhl, Bruce V. Thomsen.

**Methodology:** Nicholas J. Haley, Gordon Mitchell, Antanas Staskevicius.

**Project administration:** Nicholas J. Haley.

**Resources:** Nicholas J. Haley, Gordon Mitchell, Antanas Staskevicius, Tracy Nichols, Aaron D. Lehmkuhl, Bruce V. Thomsen.

**Software:** Nicholas J. Haley.

**Supervision:** Nicholas J. Haley.

**Validation:** Nicholas J. Haley.

**Visualization:** Nicholas J. Haley.

**Writing – original draft:** Nicholas J. Haley, Amy Buros Stein.

**Writing – review & editing:** Nicholas J. Haley, Gordon Mitchell, Antanas Staskevicius, Tracy Nichols, Aaron D. Lehmkuhl, Bruce V. Thomsen.

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
