## [Decision Letter · Decision Letter 0]

30 Oct 2019

PONE-D-19-28154

Estimating relative CWD susceptibility and disease progression in farmed whitetail deer with rare PRNP alleles

PLOS ONE

Dear Dr. Haley,

Thank you for submitting your manuscript to PLOS ONE. After careful consideration, we feel that it has merit but would benefit from the  revisions suggested by the reviewers. Therefore, we invite you to submit a revised version of the manuscript that addresses the points raised during the review process.

We would appreciate receiving your revised manuscript by Dec 14 2019 11:59PM. To enhance the reproducibility of your results, we recommend that if applicable you deposit your laboratory protocols in protocols.io, where a protocol can be assigned its own identifier (DOI) such that it can be cited independently in the future. For instructions see: http://journals.plos.org/plosone/s/submission-guidelines#loc-laboratory-protocols

We look forward to receiving your revised manuscript.

Kind regards,

Byron Caughey

Academic Editor

PLOS ONE

Journal Requirements:

1. Thank you for including your competing interests statement; "No authors have competing interests."

We note that one or more of the authors are employed by a commercial company: 'Simpson Whitetails Genetic Testing, Belleville'.

Reviewers' comments:

Reviewer's Responses to Questions

**Comments to the Author**

1. Is the manuscript technically sound, and do the data support the conclusions?

Reviewer #1: Yes

Reviewer #2: Yes

Reviewer #3: Partly

2. Has the statistical analysis been performed appropriately and rigorously? 

Reviewer #1: Yes

Reviewer #2: Yes

Reviewer #3: Yes

3. Have the authors made all data underlying the findings in their manuscript fully available?

Reviewer #1: Yes

Reviewer #2: Yes

Reviewer #3: No

4. Is the manuscript presented in an intelligible fashion and written in standard English?

Reviewer #1: Yes

Reviewer #2: Yes

Reviewer #3: Yes

5. Review Comments to the Author

Reviewer #1: Haley et. al provide data regarding PRNP genotype influence on CWD susceptibility and disease progression in whitetail deer from captive facilities. Over 2000 animals were tested in the study, finding more than 700 CWD+ cases from numerous captive deer facilities. Importantly, many rare genotypes are represented in this study. The amount of time and effort to complete such a study is commendable. Such a large test cohort was critical in making more meaningful statistical comparisons with rare alleles. Clearly showing that there are several polymorphisms that can confer higher resistance to CWD is very encouraging. I have a few minor comments that I will list below.

Lines 187-189. I think a reference to Figure 2 should be included here as that pertains to the modelling described.

Line 197. Insertion of "homozygous" between among and animals would help keep the reader clear on which genotype status is being discussed.

Table 3. In addition to the allelic frequencies provided in tables 1&2, I think it would be nice to include a column in table 3 with the actual number of deer having each listed genotype. The text often alludes to how rare some of them were, including this number would prevent the reader from having to do the math themselves (allelic frequency x the number of deer sampled).

Figure 2, upper right. Should 116GG also have a red dot for the observed outcome? Table 3 and the text in line 185 imply that such an animal was included in the study. Maybe IHC was not available for this/these deer?

Figure 2 legend. Typo. Final sentence "show" should be shown.

Reviewer #2: This paper summarizes genetic and diagnostic data from a sizable number of captive white-tailed deer herds to describe genetic influences on prion disease occurrence and progression within infected animals. As noted, the findings generally reaffirm observations and patterns reported in cited references as well as in other publications not cited here. The approach appears to be technically sound and the paper well-organized & -written.

Two corrections do need to be made:

1. The common name for this species is white-tailed deer (NOT "whitetail"). This is consistently wrong throughout the draft text, tables, and figures.

2. The term "data" is plural (for datum), and associated verbs should be corrected accordingly throughout.

Reviewer #3: This manuscript compares the Prnp genotypes of white-tailed deer with infection status to determine whether there are genotype affects susceptibility/resistance to CWD infection. The genotypes and CWD status are from farms depopulated by both the USDA and the CFIA. The authors make a number of conclusions based on this data---particularly that no genotype appears to be resistant to CWD infection, some genotypes, however, result in longer disease progressions. Uninfected herds were also analyzed to determine the genotypic variability. Of interest, 226K was found only in US white-tailed deer while 116G was found only in Canadian deer. The authors also report regional differences in allele frequencies in the captive white-tailed deer farms.

1. For the susceptibiity/resistance to CWD; it would be helpful to link CWD status in a given farm to genotype---or at least associate prevalence of CWD to CWD status for the different genotypes. It would seem likely that infection rates may overcome or, at least, impact resistance to disease---on the other hand, farms with very low infection rates may not have resulted in enough animals being infected to ensure a heterozygotes or deer that are homozygous for alleles other than 96G have been infected.

2. The authors state in the abstract that certain genotypes were not infected in some high prevalence farms. I could not find this data. This is critical information with respect to determining whether genetics play a role in susceptiblity.

3. For the regional variation in the uninfected herds---this is very interesting as the movement of cervids over the decades would likely make captive herds more similar. Are all of these samples from herds that no longer import/move deer? Was an entire herd samples, or just a subset of the animals (which could lead to a bias in the data analysis).

4. The 116G/226K data is again very intriguing---for the same reasons as mentioned above. Is there significant geographic distance between the herds analyzed in the US and those in Canada? At least with free-ranging animals, there tend not to be significant differences in allele frequencies over long geographic ranges.

6. PLOS authors have the option to publish the peer review history of their article (what does this mean?). If published, this will include your full peer review and any attached files.

Reviewer #1: No

Reviewer #2: No

Reviewer #3: No

---

## [Author Response · Author response to Decision Letter 0]

13 Nov 2019

Public Library of Science ONE

185 Berry Street, Suite 3100

San Francisco, CA 94107 USA

Dear Dr. Caughey,

 We would like to take the opportunity to thank you for your time, and that of the reviewers, in providing feedback on our manuscript entitled “Estimating relative CWD susceptibility and disease progression in farmed whitetail deer with rare PRNP alleles.” The feedback has proven very helpful in improving the manuscript, and our revisions to each comment are provided below. We first provide revisions to the Funding and Competing Interests Statements as requested:

Funding Statement:

Simpson Whitetails Genetic Testing provided support in the form of salaries for authors (DS and AC), as well as genotypic data from a subset of CWD negative herds, representing approximately 50% of the healthy herds reported. These authors had no role in study design or data analysis. The specific roles of these authors are further articulated in the “author contributions” section. 

Competing Interests Statement:

Two authors (DS and AC) were employed by Simpsons Whitetails Genetic Testing at the time this study was undertaken. These two authors provide a commercial test for PRNP genotyping to the deer and elk farming industry. There are no patents or products in development associated with this commercial venture. This association does not alter our adherence to PLOS ONE policies on sharing data and materials. 

Response to reviewers:

Reviewer #1

1. The reviewer suggests that a reference to Figure 2 be included with the statistical modelling methodology. 

Response: We have added the reference as suggested. 

2. The reviewer suggests that the qualifier “homozygous” be inserted to help clarify with genotypes are being discussed. 

Response: We have added the qualifier “homozygous” as suggested.

3. The reviewer suggests a revision to the Tables to include information on the number of deer with each listed genotype. 

Response: For this response, we chose to focus on Table 1, to include information on genotype and CWD status. We felt that this best addressed comments by reviewer 1 and reviewer 3. We have also included a supplemental Table S1 which presents genotype data for healthy herds (vs. allele frequency for Table 2). 

4. The reviewer asks whether 116GG should have an observed outcome in Figure 2; as they astutely observed, the data was present in the manuscript.

Response: The modeling data for 116GG animals is available as the reviewer notes, as shown in Table 3. The relatively low number of these animals in the study led to a fairly wide error range, outside of the scale of the graphs which were practical for the Figure (-6 log vs. -4 log range for the graphs). Because of this, the observed outcome data point for these animals (which did have IHC data available) was left out of the figure for the sake of clarity. We have provided clarification in the figure legend to point out that the data is available in Table 3. As additional data is collected on these and other animals in the future, we may be able to provide further information on how well our model fit for these graphs for homozygous animals. 

5. The reviewer identified a grammatical error in the legend for Figure 2.

Response: We have corrected this error in the final manuscript. 

Reviewer #2

1. The reviewer notes that the common name for the species reported is “white-tailed deer.” 

Response: We have amended all occurrences of “whitetail deer” to “white-tailed deer” throughout the manuscript. 

2. The reviewer points out that data is a pleural noun, and that verbs throughout the manuscript should be corrected appropriately.

Response: We have corrected the errors throughout the manuscript. 

Reviewer #3

1. The reviewer suggests that our analysis associates prevalence of CWD to CWD status for different genotypes, to address the influence of prevalence on disease resistance.

Response: The authors agree, and in fact that information is built into our modeling approach. Since this was not clear in our description of the methods, we have included clarification in our revised manuscript as follows:

“This approach accounted for differences in disease prevalence and genotypic distribution between and across farms in an effort to better estimate relative susceptibility and disease progression.” 

2. The reviewer notes that we report that some genotypes were not infected on some of the higher prevalence farms, and notes the absence of data presented which specifically illustrates this.

Response: We agree that the tables we’ve provided do not allow the reader to fully appreciate the frequency of various genotypes present – as Reviewer 1 also suggested. In response to this, we’ve modified Table 1 to include genotype data instead of allele frequency and CWD status, and we have included a supplemental Table 1 that covers genotypic data for healthy herds. 

3. The reviewer asks whether the healthy herds that were evaluated are currently importing or moving deer, and whether an entire herd was sampled for the analysis.

Response: Many of the healthy herds included in our analysis are importing and exporting deer, semen, and in some cases embryos. A small subset of these herds are entirely closed and self-sufficient. The analysis solely includes data from herds where all animals were available for genotyping, to avoid bias as the reviewer notes. We have made note of this in the methods section as follows: “Data from these healthy herds solely included locations where the entire herd was sampled”

4. The reviewer finally asks whether there was a significant distance between the Canadian and United States herds evaluated, which might offer some insight into the distribution of 116G and 226K alleles. 

Response: The geographic distance between the nearest Canadian and US herds is roughly 400 miles, however the question likely goes beyond geographical distance and more likely lies in sire selection/preferences and international laws covering the movement of deer and deer products (e.g. semen). It is worth noting that historically, the 116G allele has been reported on a farmed deer herd depopulated in Nebraska (O’Rourke et al, JGV, 2004), and this herd was rumored to be owned by a Canadian deer farmer who may have moved deer between the US and Canada prior to regulations preventing international movement of cervids. We feel that it is likely that either of these currently unreported alleles may eventually be found in their respective US or Canadian herds, however it is also likely their frequencies will be quite low. 

We submit the revised manuscript, included a copy with changes highlighted, for your further consideration for publication in PLOS One, and sincerely thank you for your assistance and contributions.

Sincerely yours,

Nicholas James Haley

Kahla Merrett

Amy Buros-Stein

Dennis Simpson

Andrew Carlton

Antanas Staskevicius

Tracy Nichols

Aaron Lehmkuhl

Bruce Thomsen

---

## [Editor Report · Decision Letter 1]

15 Nov 2019

Estimating relative CWD susceptibility and disease progression in farmed whitetail deer with rare PRNP alleles

PONE-D-19-28154R1

Dear Dr. Haley,

We are pleased to inform you that your manuscript has been judged scientifically suitable for publication and will be formally accepted for publication once it complies with all outstanding technical requirements.

With kind regards,

Byron Caughey

Academic Editor

PLOS ONE
---

## [Editor Report · Acceptance letter]

20 Nov 2019

PONE-D-19-28154R1 

Estimating relative CWD susceptibility and disease progression in farmed white-tailed deer with rare PRNP alleles 

Dear Dr. Haley:

I am pleased to inform you that your manuscript has been deemed suitable for publication in PLOS ONE. Congratulations! Your manuscript is now with our production department. 

With kind regards,

on behalf of

Dr. Byron Caughey 

Academic Editor

PLOS ONE